# Artificial Intelligence Surgery: How Do We Get to Autonomous Actions in Surgery?

**DOI:** 10.3390/s21165526

**Published:** 2021-08-17

**Authors:** Andrew A. Gumbs, Isabella Frigerio, Gaya Spolverato, Roland Croner, Alfredo Illanes, Elie Chouillard, Eyad Elyan

**Affiliations:** 1Centre Hospitalier Intercommunal de POISSY/SAINT-GERMAIN-EN-LAYE 10, Rue Champ de Gaillard, 78300 Poissy, France; chouillard@yahoo.com; 2Department of Hepato-Pancreato-Biliary Surgery, Pederzoli Hospital, 37019 Peschiera del Garda, Italy; isifrigerio@yahoo.com; 3Department of Surgical, Oncological and Gastroenterological Sciences, University of Padova, 35122 Padova, Italy; gaya.spolverato@unipd.it; 4Department of General-, Visceral-, Vascular- and Transplantation Surgery, University of Magdeburg, Haus 60a, Leipziger Str. 44, 39120 Magdeburg, Germany; roland.croner@gmail.com; 5INKA–Innovation Laboratory for Image Guided Therapy, Medical Faculty, Otto-von-Guericke University Magdeburg, 39120 Magdeburg, Germany; alfredo.illanes@med.ovgu.de; 6School of Computing, Robert Gordon University, Aberdeen AB10 7JG, UK; e.elyan@rgu.ac.uk

**Keywords:** artificial intelligence surgery, autonomous robotics, machine learning, deep learning, computer vision, natural language processing

## Abstract

Most surgeons are skeptical as to the feasibility of autonomous actions in surgery. Interestingly, many examples of autonomous actions already exist and have been around for years. Since the beginning of this millennium, the field of artificial intelligence (AI) has grown exponentially with the development of machine learning (ML), deep learning (DL), computer vision (CV) and natural language processing (NLP). All of these facets of AI will be fundamental to the development of more autonomous actions in surgery, unfortunately, only a limited number of surgeons have or seek expertise in this rapidly evolving field. As opposed to AI in medicine, AI surgery (AIS) involves autonomous movements. Fortuitously, as the field of robotics in surgery has improved, more surgeons are becoming interested in technology and the potential of autonomous actions in procedures such as interventional radiology, endoscopy and surgery. The lack of haptics, or the sensation of touch, has hindered the wider adoption of robotics by many surgeons; however, now that the true potential of robotics can be comprehended, the embracing of AI by the surgical community is more important than ever before. Although current complete surgical systems are mainly only examples of tele-manipulation, for surgeons to get to more autonomously functioning robots, haptics is perhaps not the most important aspect. If the goal is for robots to ultimately become more and more independent, perhaps research should not focus on the concept of haptics as it is perceived by humans, and the focus should be on haptics as it is perceived by robots/computers. This article will discuss aspects of ML, DL, CV and NLP as they pertain to the modern practice of surgery, with a focus on current AI issues and advances that will enable us to get to more autonomous actions in surgery. Ultimately, there may be a paradigm shift that needs to occur in the surgical community as more surgeons with expertise in AI may be needed to fully unlock the potential of AIS in a safe, efficacious and timely manner.

## 1. Introduction

Unlike artificial intelligence (AI) in medicine, which hopes to use autonomously functioning algorithms to better analyze patient data in an effort to improve patient outcomes, AI in surgery also involves movement. Unlike strict medicine, surgery is an art that is also dynamic. When we use the term “surgery” we are also referring to endoscopy and interventional techniques and procedures, because interventional disciplines continue to coalesce into the same field, a trend that is seen by the continued increase in hybrid operating rooms that have angiography compatible tables, mobile CT scanners, minimally invasive surgical equipment and endoscopes all in the same room that can be used in tandem. Because of the fact that interventional fields of medicine also rely greatly on the medical management of patients, we believe that AI medicine (AIM) and AI surgery (AIS) could one day be considered two distinct disciplines, with AIM reserved for instances where computer algorithms are used to better diagnose, manage or treat patients without a specific interventional procedure being done.

AIS could be a term for autonomously acting machines that can do interventional gestures/actions. According to the Gartner Hype Cycle, many surgeons believe that we are languishing in the “Trough of Disillusionment”, and the promise of autonomous surgery seems like a “pipe dream” for most modern-day surgeons [1,2]. However, the reality is that instances of autonomous actions in surgery already exist. Unfortunately, the reluctance of many laparoscopic surgeons to give up on haptics, or the sense of touch, is actually hindering progress in AIS because of the refusal to embrace robotic tele-manipulation technology, in effect, they are refusing to let go, something that will be needed if the dream of AIS is ever to come to pass [3]. Unfortunately, the medical community has already been shown to be resistant to any automation of medical tasks even simple computations. It is safe to say that automation of surgical tasks will have an even more profound degree of resistance [4].

Another obstacle to the growth of AIS is the dogmatic belief that for something to have AI, it must have algorithms that enable progressive improvement and learning by an artificially intelligent device [5,6]. This creates a conundrum, as theoretically, machine learning (ML) should be infinite, and because of this one wonders what the ultimate purpose of perpetual learning is in surgery. Should it be carried out in the hopes that ultimately the surgical action will become so perfect that it is no longer necessary? How much more perfectly does a robot need to place a titanium clip on the cystic duct?

It could be argued that technology used to create monopolar and bipolar technology is an example of AIS as it has tissue sensing technology that adjusts the action of cautery based on the resistance and capacitance of the tissue (TissueFect™, ValleyLab, Medtronic, Dublin, Ireland). In particular, it has a closed-loop control that analyzes 434,000 data points per second. Does this technology need to improve on that level of data analysis to be considered AI? Or what about Ligasure technology, which uses a high-speed advanced algorithm to seal vessels with changes to the duration of action dependent on tissue thickness (Ligasure, ValleyLab, Medtronic, Dublin, Ireland). We certainly do not mean to imply that there is no room for improvement in these technologies, but at what point should something be defined as AI? Shouldn’t any autonomous action be acknowledged and celebrated as an example of AIS?

ML, Deep Learning (DP), Computer Vision (CV) and possibly Natural Language Processing (NLP) will ultimately be the best path towards more autonomous actions in surgery, but in the meantime, handheld robotics and complete robotic surgical systems are a necessary step that modern day minimally invasive surgeons will need to embrace to ultimately realize the dream of autonomously functioning robots in surgery. This manuscript will discuss the literature on ML, DL, CV and NLP as it pertains to AIM for the pre-operative diagnosis and post-operative management of surgical patients and autonomous robotics in surgery, to attempt to ascertain the current obstacles and necessary next steps in the evolution towards AI/autonomous surgery.

## 2. Machine Learning

Machine learning (ML) is a genre of artificial intelligence including algorithms that allow machines to solve problems without specific computer programing. While analyzing big data, machines are enabled to assimilate a large amount of information, applicable for risk stratification, diagnosis, treatment decisions, and survival predictions. Not only can AI models analyze large amounts of data collected over long periods of time, providing predictions for future events on the basis of the statistical weight of past correlations, they can also continuously improve with new data. Through a process called “incremental learning”, trainable neural networks improve over time, surpassing unchanging scoring systems and standardized software. Moreover, the human–machine interaction further improves the performance of ML tools. Indeed, the learning process goes far beyond the textbook data, incorporating real-life scenarios and can improve experts’ opinions.

Most of the studies conducted on ML tools have focused on machine vision, biomarker discovery, and clinical matching for diagnosis, classification and outcome prediction [7]. Several studies have applied different ML tools to surgery and, in particular, to risk assessment, performance evaluation, treatment decision making and outcome prediction. In an effort to better identify high-risk surgical patients from complex data, a ML project trained on Pythia was built by Corey et al. to predict postoperative complication risk [8]. By using surgical patient electronic health record (EHR) data, including patient demographics, smoking status, medications, co-morbidities, surgical details, and variables addressing surgical complexity, the authors created a decision support tool for the identification of high-risk patients. Similarly, Bertsimas et al. applied novel ML techniques to design an interactive calculator for emergency surgery [9]. By using data of the American College of Surgeons National Surgical Quality Improvement Program (ACS-NSQIP) database, the authors designed the POTTER application (Predictive OpTimal Trees in Emergency Surgery Risk), to predict postoperative mortality and morbidity [9]. Differently from the standard predictive models, POTTER accounted for nonlinear interactions among variables; thus, the system reboots after each answer, interactively dictating the subsequent question. A similar tool, predicting eight specific postoperative complications and death within one year after major surgery, was developed and validated by Bihorac et al. [10]. MySurgeryRisk can accurately predict acute kidney injury, sepsis, venous thromboembolism, intensive care unit admission >48 h, mechanical ventilation >48 h, and wound, neurologic, and cardiovascular complications with AUC values ranging between 0.82 and 0.94 and death up to 24 months after surgery with AUC values ranging between 0.77 and 0.83. The following studies built different ML models predicting postoperative outcomes that proved to perform better than traditional risk calculators [11,12].

In an attempt to better define patient outcomes, Hung et al. applied ML to objectively measure surgeon performance in a pilot study [13]. By combining procedure-specific automated performance metrics with ML algorithms the authors were able to objectively assess surgeon’s performance after robotic surgery. These algorithms still include biases and misinformation, and thus multi-institutional data from several high-volume groups are needed to train the tool and to create a robust model that is able to correctly predict different clinical outcomes.

Surgical performance has also been measured to allow surgeons to learn from their experience and refine their skills. Either by using surgical video clips or by applying virtual reality scenarios, deep learning models were trained to estimate performance level and specific surgical skills [14,15]. In the near future, we will be able to personalize training through ML tools such as these.

Other studies have used ML for prediction of staging [16], for treatment decision making [17], and to improve case duration estimations [18], further expanding the applicability of ML. Despite these advantages, ML presents several challenges, such as the need to process a large amount of data before it can be analyzed, the necessity of repetitively training the model, and of refining it according to the various clinical scenarios. Ethical considerations should also be taken into account when applying ML to healthcare, including privacy and security issues, and the risk for medico-legal implications. These issues will be discussed at more length below [7].

## 3. Natural Language Processing

Natural Language Processing (NLP) is the evolution of the interaction of Artificial Intelligence (AI) and linguistics. Distinct from simple text information retrieval (IR), which indexes and searches a large volume of text based on a statistical technique [19], NLP evolved from basic approaches (word to word), through a complex process of coding words, sentences, meanings and contexts, to its modern structure. Since 1950, NLP and IR converged into what is known today as NLP, namely, a computer-based algorithm that handles and elaborates natural (human) language, making it suitable for computation [20].

When applied to healthcare, where available clinical data are kept in Electronic Health Records (EHRs), the need to decode a narrative text coming from this large amount of unstructured data has become urgent because of the complexity of the human language and the routine employment of metaphors and telegraphic prose. When compared to NLP, manual reviewing of EHRs is time consuming, possibly misleading because of biases, and extremely expensive [21].

The tremendous potential value of a big data analytical system in healthcare can be easily explained: EHRs represent at this time the major source of patient information, but unfortunately, for the most part, data regarding symptoms, risk factors for a specific disease or outcomes after medical or surgical treatment come from unstructured text. The ability to translate this amount of information into a coded algorithm could allow for more precise screening programs and modify medical and/or surgical strategies. A systematic review from Kolech et al. analyzed the available methods, employing NLP to interpret symptoms from EHRs of inpatients and outpatients, finding possible future applications for NLP in the normalization of symptoms to controlled vocabularies, in order to avoid overlapping of different words for the same concept [21]. A notable criticism of the available studies has been that reported signs and symptoms are easily mixed as the same variable, making interpretation confusing. In this review, only 11% of studies focused on cancer patients, in contrast with the fact that, currently, a major area of interest for AI (not only NLP) is oncology, where early detection of cancer-specific symptoms could facilitate early diagnosis and potentially enhance screening techniques.

An obvious and immediate advantage of having reliable and decoded data coming from clinical notes is the positive impact on the quality of retrospective studies. Moreover, NLP analysis of symptoms and signs in cancer patients may allow for the improved definition of prognostic factors other than surgical and oncological parameters [22]. Emotional status and quality of life of patients after cancer surgery or other cancer treatment has also been investigated through NLP [23,24]. Banerjee et al., with the creation of a specific domain vocabulary of side-effects after prostatectomy, were able to evaluate minor outcomes hidden in clinical free text, resulting in better management, which could be a game-changer in a population with a 5-year life expectancy rate approaching 99% [23].

When applied to surgery, NPL has been extensively proposed pre-operatively and looking at different post-operative complications such as surgical site infection (SSI). Bucher et al. developed and validated a model of SSI prediction with NLP algorithm by analyzing EHRs from 21,000 patients entered into the ACS-NSQIP, using only text-based documents regarding history and physical condition, operative, progress and nursing notes, radiology reports and discharge summaries [25]. This predictive model had a sensitivity of 79% and a specificity of 92% on external validation, but its added value was the absolute reliable negative predictive value (NPV), which is a relevant issue for events with a low incidence. Anastomotic leak [26], deep venous thrombosis, myocardial infarction and pulmonary embolism were also frequently investigated and results from a recent meta-analysis [27] demonstrated that performance of NLP methods in detection of postoperative complications is similar, if not superior, to non-NLP methods, with a sensitivity of 0.92 versus 0.58 (*p* < 0.001), and comparable specificity. Moreover, NLP models seem to be better than non-NPL models for ruling out specific surgical outcomes, owing to an optimal true-negative identification power. Interestingly, the ability of algorithms to self-correct can increase the utility of their predictions as datasets grow to become more representative of a patient population.

These NLP applications are surely beneficial for patient management, providing better understanding of peri-operative data, but it can be a useful tool for surgeons as well, particularly when applied to surgical education. For example, decoding intra-operative dialogues between residents and faculty, combining NLP and CV, can create and implement a dataset of technique and surgical strategy, thus, creating a real-life textbook of surgery. In addition, NLP has been efficiently used [28] to assess Entrustable Professional Activities (EPAs). EPAs describe specific behaviors associated with different training levels of residents and it can potentially enhance the understanding of their training and autonomy in surgical practice.

NLP can be used to validate datasets that are the basis of surgical risk predictive models, but the main limit of their widespread use is the non-homogeneity of NLP models and EHRs data entry forms across institutions and countries. A future improvement would entail expansion of registries from local to national and international levels to set algorithms that can be externally validated on various populations. Surgeons and their low-level confidence with AI represent another limit to developing a system that is theoretically perfect and promising: it is important for them to understand how AI may impact healthcare and to elaborate strategies of safe interaction to implement this nascent technology. Synergy between fields of AI is also essential in expanding its applications. Lastly, ethical issues and privacy rules protecting patients’ sensitive data, can limit the large-scale applicability of NLP over EHRs. Nonetheless, the enormous potential of NLP remains fascinating and the multiple potential benefits of its integration into healthcare must be balanced with risks. Although the technology does not currently exist for NLP to influence autonomous actions in surgery, it must be remembered that communication among team members during surgery is fundamental to the successful performance of surgery. Additionally, devices already exist and are used today that are voice-activated (ViKY, Endocontrol, Grenoble, France), and it is conceivable that NLP could eventually evolve to benefit the action of voice-controlled devices during a procedure [29,30].

## 4. Deep Learning and Computer Vision

Core Computer Vision (CV) tasks include the development of methods to process and analyze digital images and video streams to localize, recognize and track objects of interest [31]. These tasks are considered cornerstones in most autonomously or semi-autonomously acting machines and applications (e.g., robots and self-driving cars) and indeed are critical in AIS and medical image analysis [32]. A typical vision task such as localizing and recognizing objects of interest (e.g., critical findings in chest X-ray images) [33] can be approached either using traditional vision, or advanced deep-learning-based methods (Figure 1).

### 4.1. Traditional vs. Deep Models

Traditional CV algorithms were based on extracting a set of low-level or high-level features from images or videos (e.g., points of interest, color intensity, edges, etc.), and then using these features to train a supervised learning model such as a support vector machine (SVM) [34], random forest (RF) [35], or other models, to recognize an object of interest or classify an image. Over the past four decades, various methods have been developed to extract key discriminating features from images, including edge detectors [36], scale-invariant features such as SIFT and SURF [37,38], and others, which have helped push the research boundaries in CV as a whole. Despite this, it can be argued that before the widespread development of DL and deep convolutional neural networks (CNN) [39] in 2012, advances in CV were marginal, and even simple vision tasks were considered inherently challenging. One key problem of traditional vision methods is that they rely almost completely on the quality of the extracted features, and in many scenarios, such as AIS, it is difficult to know what features should be considered and then extracted to perform specific tasks. As can be seen in Figure 1, poor features would essentially lead to poor results, regardless of how complicated the learning models are. Furthermore, most of these traditional methods are domain dependent, and are sensitive to noise, light conditions, and the orientation of the objects in the images.

Unlike the aforementioned traditional methods, deep learning methods learn the underlying representation of the image in an end-to-end manner and without the need to handcraft these features [39]. These methods have revolutionized the application of AI across various domains, and significantly improved performance by orders of magnitude in many areas, such as in gaming and AI [40], NLP [41], health [42], medical image analysis [33], and cyber security [43], among others. As can be seen in the schematic diagram in Figure 2 (below), the key advantage of DL methods is the ability to map a set of raw pixels in an image to a particular outcome (e.g., an object’s identity, abnormality) [39], which proved to be inherently challenging tasks for traditional CV methods.

### 4.2. Advances in Computer Vision

CNN-based methods, in particular, have made significant progress in recent years in the CV domain [44]. They have been successfully applied in several fields such as hand-written recognition [45], image classification [46,47], and face recognition and biometrics [48], among others. Prior to CNNs, the improvements in image classification, segmentation and object detection were marginal and incremental. However, the introduction of CNNs revolutionized this field. For example, Deep Face [49], a face recognition system first proposed by FaceBook in 2014, achieved an accuracy of 97.35%, beating the then state-of-the-art methods by 27%.

Core CV tasks that are considered a cornerstone for any serious AIS attempt [42] such as shape and object detection, recognition, and tracking have become much less challenging even under different conditions and in much less controlled environments. Object detection and recognition have always been the main challenge for traditional CV methods. However, faster region-based CNN (R-CNN) [50], single shot detectors (SSD) [51], region-based fully convolutional networks (R-FCN) [52], and YOLO (you only look once) [53] are all relatively recent methods that have shown superior performance in the field of object detection, tracking, and classification. These methods and their extensions have significantly advanced this area of research and solved some of the most challenging and inherent vision problems, such as occlusions, light conditions, and orientation, among others, which were considered major challenges, even for a specific vision task in a more controlled environment [54]. Additionally, they provide a unique opportunity to advance research and development in AIS and robotic-assisted surgery.

These newly developed methods for localizing and recognizing objects of interest from images have resulted in significant and unprecedented progress in the field of medical image analysis and understanding [55]. For example, CNN-based methods continue to play a key role in advancing classification tasks in the medical domain and providing tools for supporting decision making in clinical settings. Pomponiu et al. [56] used a pre-trained CNN to learn the underlying representation of images and extract representative features that help detect skin cancer. Then, they fed the extracted features into a *k* nearest neighbor classifier (kNN) and reported an average accuracy of 93% with similar sensitivity and specificity. Similarly, Esteva et al. [57] trained a CNN model using a dataset of 129,450 clinical images representing various skin diseases. The authors also used a pre-trained model to boost performance and reported results comparable with experts. It should be noted that the use of pre-trained models in such tasks is very common and referred to as Transfer Learning (TL).

The key motivation behind such an approach is the fact that training deep models requires large volumes of data (e.g., tens of thousands of images), which may not always be readily available, and can be very expensive to collect and annotate. Therefore, transfer learning provides an opportunity to take advantage of deep models that have already been trained on millions of images. Typical models that are commonly used include AlexNet [47], VGG-16 [58], and GoogleNet Inception [59], among others. A common theme among these models is that they are trained on a large number of images (more than 1.28 million images), and have a deep architecture that enables them to classify 100′s of different objects. In other words, these models are designed to solve very complex tasks for object detection and classification from large volumes of data, and training requires powerful machines and long hours.

TL proved to be very useful in the application of CNN-based methods to various tasks related to cancer detection and diagnosis. A thorough and recent review clearly shows how CNN-based methods, along with pre-trained models, have advanced research in many areas, such as breast, prostate, lung, skin and other types of cancer [60]. Similarly, Levine et al. [61], in a relatively recent review, showed how DL helped to greatly improve cancer diagnosis, and argued that in particular radiology- and pathology-related tasks, DL-based methods achieved comparable performance to medical experts.

CNN-based methods have also greatly advanced research in the area of object localization. This is an important and core task in medical image analysis, by which the machine not only learns to classify an image as normal or abnormal, but also to localize and highlight the area in the image that exhibits the abnormality, if it exists. Examples include the processing and analysis of magnetic resonance imaging (MRI) to detect brain tumors [62], segmenting CT scan images [63], and localizing the area of interest from chest X-Ray images [33]. Such tasks were inherently challenging for conventional ML methods, which requires the careful design of heuristics. However, with DL it became possible to train an end-to-end solution to solve such complex tasks. For instance, Schwab et al. presented a new approach for localizing and recognizing critical findings in chest X-ray images (CHR), using a multi-instance learning method (MIL) that combines classification and detection [33].

MIL is based on dividing the image into a set of patches (patch-based convolutional neural network), and using the prior knowledge of positive and negative images while the proposed method learns which patches in the positive images are negative [33]. Three different public datasets were used, and competitive results were reported. Similarly, Schlemper et al. [63] presented a method incorporating Attention Gate (AG), a mechanism used to help identify salient image regions, into CNN architecture to learn and highlight arbitrary shapes and structures in medical images, and used it for classification and object localization tasks, where promising results were reported in terms of overall prediction performance and segmentation of 3D CT abdominal images.

This progress and advancement in solving such complex vision tasks can be attributed to three main factors. First, the rapid development of the underlying algorithms based on deep neural networks (DNNs) and deep CNNs [39]. Second, the computing power, and the availability of Graphics Processing Unit (GPU)-based machines and cloud services have made it possible to train very deep models using large volumes of images and unstructured documents. The third reason for this progress in CV is the availability of public medical image datasets, helping to accelerate research and development in this field. Examples include datasets related to musculoskeletal radiographs such as MURA [64], which contains 40,561 images from 14,863 studies representing 11,184 different patients. Another example is the colon cancer screening dataset [49], containing 40 colonoscopy videos and containing almost 40,000 image frames, lung image datasets [65], and others. However, more importantly, today there are various online platforms that provide access to similar datasets and provide online computing services to enable researchers to build, test, and share their results. One of the most important platforms is Kaggle (https://www.kaggle.com/ accessed on 16 August 2021), which hosts various types of medical datasets.

### 4.3. CV in AIS

The significant progress that occurred in object recognition and localization in 2D and 3D images has been reflected in autonomous surgery across different types of applications, including phase recognition [15,66,67], detection and tracking of objects of interest [68,69,70], and segmentation [71,72]. Phase recognition is an important aspect for the training and education of doctors using videos of various types of surgery. However, despite the availability of these videos, their use in training is still limited, because these videos require some sort of pre-processing, and also the segmentation into different phases for subsequent automated skill assessment and feedback [15].

To address this issue, Twinanda et al. [66] built a CNN-based method to perform phase recognition in laparoscopic surgery directly from raw pixels (image and videos). The authors used a dataset of 80 videos of cholecystectomies performed by 13 surgeons to train the CNN, and promising results were reported in terms of the model’s ability to handle complex scenarios and outperform other traditional tools. Similarly, Yu et al. [67] used five different algorithms including CNN for handling the of videos of cataract surgery and Recurrent Neural Networks (RNN) for handling time series data with labels. The results clearly showed that deep learning techniques (CNN and RNN) provide better options for learning from time series data and video images, and can provide accurate and automated detection of phases in cataract surgery. Khalid et al. [15] used deep convolutional neural networks with embedding representation for phase recognition. The authors used 103 video clips of table-top surgical procedures, performed by eight surgeons, including four to five trials of three surgical actions. Promising results with respect to precision, recall and accuracy were reported in terms of the model’s ability to classify performance level and surgical actions.

Object detection and tracking is another important aspect of AI in surgery that has progressed due to the latest developments in deep learning and deep convolutional neural networks. Sarikaya et al. [69] used a dataset of videos from 10 surgeons and applied a deep convolutional neural network to speed up detection and localization of instruments in robot-assisted surgery (RAS). The authors used multimodal CNNs to capture objects of interest and the temporal aspect of the data (motion cues). Results with 91% precision were reported, along with relatively good performance in terms of computational time (0.1 s per frame).

Tracking of objects of interest across different frames is another key aspect of AIS that has also been advanced due to the latest developments in computer vision. Lee et al. [70] proposed a deep-learning-based method for tracking surgical instruments to evaluate a surgeon’s skills in performing procedures using robotic surgery. The authors used 54 videos to train their models and used mean square root error and the area under the curve for evaluation purposes. The results showed that the proposed method was able to accurately track instruments during robotic surgery. The authors concluded that the results suggest that the current method of surgical skill assessment by surgeons could be replaced by the proposed method.

One particular application of CV that has seen significant progress in recent years due to developments of deep-learning-based methods is image and video segmentation. Accurate segmentation of images and videos is crucial for AIS and robot-assisted surgery. A notable example from the Massachusetts Institute of Technology proposed a deep-learning-based approach for robotic instrument segmentation [72]. The authors proposed an architecture based on deep residual models (U-Net) [73]. The method presented provides pixel-level segmentation, where each pixel in the image/video is labeled as an instrument or background, and the authors used eight videos (each one of 255 frames) to train their models, and reported comparable results to those obtained with state-of-the-art methods. Similarly, in [71], the authors used fully convolutional network and optical tracking for segmentation of computer-assisted surgery videos. Overall results of 80.6% for balanced accuracy were reported in a non-real-time version of the method, dropping to 78.2% balanced accuracy in the real-time version.

Various other applications of CV methods and AI can be seen in AIS, including application of CV and AI for education in surgery [74], to improve efficiency in the operating room [75], during neurosurgery [76], and in other surgical disciplines. For a comprehensive and recent review of the use of computer-vision-based methods in AIS and assisted surgery, the reader is referred to [77].

### 4.4. Reinforcement Learning

Reinforcement learning (RL) is a branch of ML that uses Markov Decision Processes (MDP), which are based on Markov chains, named after Andrey Markov, a Russian mathematician who used this framework to help predict the outcomes of random events [78,79]. In addition to AI in medicine, it has also been applied to autonomous actions in surgery [80]. As noted, enabling the computer to see and recognize things was, in the past, a great hindrance to advancements in AI. Ultimately, an algorithm using a fusion of kinematic and video data was created based on Markov models that enabled segmentation of distinct surgical movements [81]. Combining these innovations with neural networks of DL has enabled vast improvements in the reaction time and overall speed of these complex gestures, and is known as deep reinforcement learning [81].

Currently, most research in RL and surgery involves simulations [82]. Due to the vast amount of data that must be analyzed, researchers first used 2-dimensional (2D) models to devise researchable RL applications. One such model involves a 2D model for teaching gynecological surgeons how to do a hysterectomy. The researchers theorized that by defining precise goals or rewards (also known as “models”), and reproducible tasks or movements mimicking a hysterectomy, that they could combine RL with surgery and develop worthwhile algorithms that could be analyzed, studied and improved [82].

Subsequent teams developed algorithms that were able to control the robotic arms of the da Vinci system and complete tasks in three dimensions (3D), specifically, to identify a real 3D cylinder with precision and accuracy and move the 3D object [83]. Initial work was performed using a simulation and then transferred to real life. They also used MDPs to accomplish these tasks. Ultimately, these researchers were able to program the robot to identify fake blood and shrapnel and to remove this from real life box trainers. These researchers ultimately made this technology open source and called it dVRL, or da Vinci Reinforcement Learning [83].

Other groups also used RL with the da Vinci robot simulator, but with reference to model-free RL algorithms, or non-MDP algorithms [84]. A simple way to understand model-free RL is to think of it as learning based on “trial and error”, as opposed to other approaches that are based on “rewards” or, as mentioned above, “models”. A group from Johns Hopkins termed their non-model algorithm Hybrid Batch Learning (HBL) and found that they could improve the speed of their robot using this approach. They added a second replay buffer to a Q-learning process that used interpreted data from recorded images to streamline decisions on movements, resulting in reduced times for autonomous robotic actions. Q-learning is a form of model-free RL, and may be the most promising form of RL for autonomous actions in surgery as it most closely resembles the human experience [84]. A Q-learning form of RL was also recently developed using the da Vinci Research Kit (Intuitive Surgical, Sunnyvale, CA, USA) to create a system to perform autonomous surgical knots using the robot [85]. However, as in current surgery, what we gain in speed we often end up paying for in increased complications, which brings up the important question of ethics in AI surgical innovation.

Imitation Learning (IL) can be considered an emerging field from the domain of RL, and is a result of recent advances in deep learning and deep convolutional neural networks. The idea is based on teaching an agent (a learning machine/algorithm) to perform a certain task through demonstrations. The overall aim is to learn a mapping function h(x) that creates a map between observations and actions [86]. IL-based methods have been applied successfully to generic navigation tasks, where the agent learns from visual observations (mostly videos or images) [87]. In a recent study, Kim et al. used reinforcement learning to facilitate accurate manipulation and navigation of surgical tools during eye surgery; in particular, they proposed a method to predict the relative goal position on the retinal surface from the current tool-tip position [88].

### 4.5. Challenges in AIS

Although CV tasks play a crucial role in AIS, autonomous surgery is way more complex than computer vision. It involves movement, navigation, object recognition, classification, taking actions, and much more. Each of these components is a challenging problem by itself, and this can explain why having fully autonomously acting machines in surgery is still far from a reality. That said, the development in CV tasks has made it possible to provide some form of robot-assisted surgery that can help in decision making and acting processes [89]. From a computer vision perspective, there are several challenges that can be considered to be barriers to the widespread development of AIS and robot-assisted surgery. However, two key challenges include data availability and data annotation.

#### 4.5.1. Data Availability

Although various datasets are available in the public domain (as discussed earlier), a very small number of these datasets are designed for autonomous or semi-autonomous surgery. Such datasets are very important for advancing research in AIS and semi-autonomous surgeries [90]. An example of an attempt to prepare a dedicated dataset for AIS is the work presented in [91], which presented an experimental framework to evaluate the performance of CV methods in analyzing operative videos to identify steps of a laparoscopic sleeve gastrectomy (LSG). The authors collected videos capturing operations from start to finish for 18+ patients, and using DNNs, they reported an overall accuracy of 85.6% in identifying operative steps in laparoscopic sleeve gastrectomy, such as port placement, retraction of the liver, liver biopsy, etc.

#### 4.5.2. Data Annotation

The second challenge, which is also closely related to data, is the data annotation process. The annotation process can be defined as simply the process of allocating and annotating the areas of interest in a video or an image. For example, consider a classification task that aims at learning whether an X-ray contains an abnormality or not. This requires experts to label large volumes of such images as either positive or negative instances. Such tasks, however, become more complex when we are talking about defining and allocating areas of interest in a video that captures a full surgical procedure. This process is expensive, time consuming and largely depends on a human’s expertise. Moreover, the annotation process is performed almost manually, where the users are required to manually allocate these areas of interest, often using open-source tools that capture the location and boundaries of these areas.

#### 4.5.3. Ethics of Technological Advancements in AIS

The ethics of the introduction of new surgical devices is a complex issue that has been divided into four broad principles: beneficence, non-maleficence, autonomy and justice [92]. In general, six groups of people are considered when discussing this issue: the medical regulatory board, surgeons, patients, hospitals and surgical centers, industry and surgical societies [93]. The issue of ethics as it pertains to artificial intelligence surgery or more specifically autonomous actions in surgery is unclear because in general surgical robots that have already been approved for use in humans will ultimately be the vehicle used to deliver the artificially intelligent movements. Technically, the artificial intelligence will not be inside of the patient, but the robotic arms will be. Approval for technology that does not enter into the patient is generally easier to obtain then for new technology that goes inside patients. Current examples of autonomously functioning devices in patients include, for example, automatic staplers (iDrive, Medtronic, Dublin, Ireland). As mentioned, this device has a sensor and changes the speed of stapling and functions autonomously once activated. As autonomous actions become more complex, it begs the question of whether or not approval of more artificially intelligent devices will require a more rigorous approval process, even if the AI technical remains outside of patients.

As opposed to the ethical issues of AI in the collection, manipulation and interpretation of medical data, AIS has the added potential danger of real-time analysis of intra-operative issues and potential for complications [94]. Alternatively, it could be argued that AI may result in fewer complications because of technology devised to minimize complications. Clearly, we are many years away from being able to truly study this; nonetheless, it is clear that more surgeons need to become well-versed with issues of AI so that surgeons can truly partner with engineers and computer scientists in the development of safe autonomous actions.

Currently, four classes of medical devices based on risk have been designated. Class 1 devices have minimal risk to the patient and remain in limited contact with the patient and include surgical instruments; class 2 devices include things like CT scanners, ultrasound probes and contact lenses; class 3 includes hemodialysis and implants; and class 4 includes implantable defibrillators and pacemakers [95]. One wonders if a fifth class should be designated for devices that make real-time clinical decisions using AI during procedures. The consequences of not having surgeons be intimately knowledgeable and involved in the development of new technologies such as surgical robotics and AI have been divided into five ethical categories of experience for surgeons and additionally patient expectations and include: rescue, proximity, ordeal, aftermath and presence [96]. It is in this spirit of humility and acknowledgement of the fundamental role of morality in surgery that this article was written.

## 5. Discussion

All fully autonomous robots have elements of ML, DL, CV and NLP. Although so-called “Strong” AI does not yet exist in surgical robots, examples of “Weak” AI do exist and permit the independent performance of automatic linear stapled gastrointestinal anastomoses and robotic arm alignment during operating table tilting [2,6]. Another way to define Weak AI in surgical robotics is to consider things like the iDrive and da Vinci/Wolf table as reactive agents only as opposed to deliberative agents. Deliberative agents are the hope for Strong AI and would include information gathering, knowledge improvement, exploration, learning and full autonomy. Reactive agents are also known as “finite automata”, and only carry out simple tasks. Essentially, “Weak” AI performs an automatic action, but without the idealized concept of intelligence that one would expect in AI, specifically, a complex interpretation of a problem with an advanced solution equal or superior to a humans. Nonetheless, these examples of autonomous actions are the small steps that we need on the road towards more complete automation. Some researchers in AI believe that AI must ultimately include computers with desires, beliefs and intentions. Although an automated surgical robot should ultimately have the ability to act with intention, it is hard to see the need for a robot having beliefs or desires to function effectively during an interventional procedure; however, this may not be the case in the future once AI is much more advanced [97].

In accordance with this, an international group of robotic surgical experts recently defined six levels of surgical autonomy: level 0 designates no autonomy, level 1 is defined as the tele-manipulation of a robot, level 2 corresponds to a limited autonomous action; level 3 is defined as conditional autonomy where the surgeon selects among several autonomous actions that the robot can perform; level 4 autonomous actions include automated medical decision making under a doctor’s control; and level 5 indicates full autonomy where no human control is necessary. It would appear that the complete surgical system as it is known today fits level 1, but vessel sealing devices such as the Ligasure or cautery devices would be level 2, and automatic stapling devices that have internal sensors could arguably be described as meeting the criteria of level 3. Interestingly, a team from Germany modified a previously developed robotically controlled laparoscope holder [29] to move autonomously and improve with time, resulting in decreased laparoscopic cholecystectomy times [98]. This device could indicate that we have already entered level 4 in humans [98].

In the clinical world, ML and DL are also currently limited to off-line data interpretation such as in Radiomics and evaluation of surgical performance [15,99]. The power of DL is elucidated by the observation that convolutional neural networks have already been shown to function at the level of average breast radiologists in differentiating radiographic breast findings [100]. CV, also known as robotic vision, also relies on DL. DL is a form of ML that creates models from large and complex data sets; however, ML can also create models from sets of data that are not pre-labeled which is known as Unsupervised Learning and models that use algorithms to interact with the environment in real time, which is known as Reinforcement Learning. Ultimately, more autonomous robotic actions will require superior methods of Unsupervised and Reinforcement Learning [101]. The shear amount of data necessary to interpret the vast amount of data generated from robotics in real time is hard to fathom. The hope is that NLP may ultimately be able to facilitate this interaction with man, robot and computer. However, one has to wonder if there is one fundamental flaw in the way that surgical robots are being designed.

Although robotic autonomy has always been the driving factor in even the earliest surgical robots [102,103], a counter-intuitive push has existed for the development of haptics or the perception of touch [104,105]. Measurement of the force exerted on robotic tips focuses on measurements obtained from the tip of robotic instruments. Designs have ranged from two-degree-of-freedom force readings via torsional springs and flexure hinges to graspers with four-degree-of-freedom readings [106,107]. More sensitive robotic tip sensors have also been developed with polyvinylidene fluoride (PVDF) that have the strength to detect force measurements of graspers, but which also detect pulsations of arteries [108]. The fact that the sensors are at the tip of the instrument leads to a whole host of problematic issues from size constraints of components to limitations involving need for repetitive sterilization.

If the goal of robotic surgery is full automation, and all surgeons who use the complete surgical systems operating today are able to perform surgery without haptics, relying on visual cues alone, one has to wonder if the development of haptics is truly necessary. A CNN-based framework to autonomously control guide wires in vessels during angiography is currently being developed that will use haptics recorded during robotic angiography performed by a clinician to develop algorithms that can enable an autonomously functioning robot to successfully navigate vessels intra-luminally [109]. As with humans, the current models involve two-dimensional image interpretation with fluoroscopy being converted into three-dimensional robotic movements [110,111]. A key aspect of this process is that although the robot will be trained with haptic information, ultimately the procedure will be performed without the need for the clinician to sense and interpret the haptic information. This indicates that although haptics may be fundamental to developing these movement algorithms, ultimately haptics interpreted by humans may no longer be necessary.

Interestingly, from the very beginning of surgical robotics, researchers noticed a problem with “noise” and ultimately had to use filters to obtain interpretable data [102]. What if the robot did not need to feel like humans, and what if the noise is what we should be focusing on? Interestingly, the Versius Complete Surgical System has haptic capabilities, but the detection of resting human tremor makes the haptics useless and potentially bothersome [3]. Symbolically, one of the first tools in modern medicine was sound, and, as highlighted by the development of the stethoscope, it is in this spirit that researchers began to harness the information generated by friction, bumps and perforations by analyzing them with various ML algorithms, finding that differentiation of these events was possible during vascular catheterizations with guidewires [112]. The ML algorithms included artificial neural networks, K-nearest Neighbor and Support Vector Machine. Unlike the sensors of robotic graspers, however, this sensor was able to be placed on the proximal end of the guide wire and not on the distal part in the patient [113,114].

Interestingly, this technology has been shown to have the ability to generate quantifiable information from robotic graspers, while the sensor itself still remains outside of the patient [115,116,117]. The potential of this technology has also been shown by demonstrating that everyday procedures like needle insertions can become smarter and potentially safer, specifically in the case of Veress insertion for pneumoperitoneum access, liver ablation of tumors and arthroscopic insertions [118,119].

## 6. Conclusions

The complexity of the field of artificial intelligence surgery is highlighted by the finding that only 9.8% of medical devices undergo in-human testing within 10 years, and that involvement with an actual clinician during the development of new devices significantly improved the chances of this [120]. Advancing AIS requires compiling large volumes of videos that captures surgery procedures. This requires hundreds if not thousands of fully annotated videos for each specific type of surgery that can be used and shared within the research community. To meet this requirement, data collection, preparation and annotation must be part of future medical practice. It also requires close and cross-disciplinary collaboration from the AI and medical communities. Autonomous actions in robotic surgery will involve a complex interplay of ML, DL, CV and potentially NLP. Modern surgeons must become versed with the basics of AI to better incorporate this exciting new field into surgical practice. Young academic surgeons should consider gaining experience in this field in the form of Masters or PhD programs, as opposed to more traditional fields of study such as molecular biology, genetics and immunology.

## Figures and Tables

**Figure 1 sensors-21-05526-f001:**
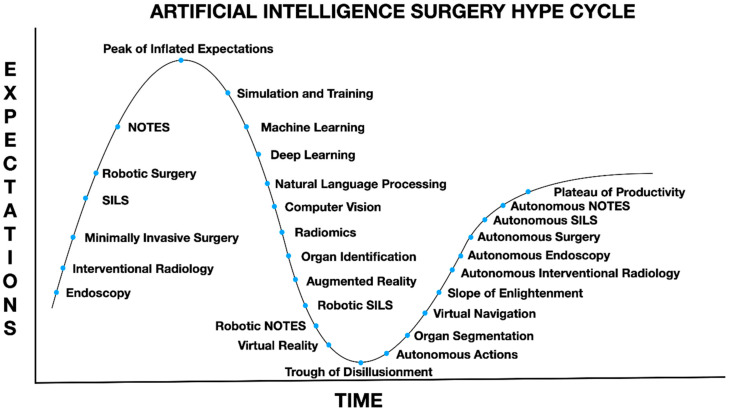
Artificial intelligence surgery Gartner hype cycle: natural orifice transluminal endoscopic surgery (NOTES), single incision laparoscopic surgery (SILS), machine learning (ML), deep learning (DL), natural language processing, computer vision (CV). Adapted from the Gartner Hype Cycle for Artificial Intelligence, 2019 gartner.com/smarterwithgartner and modifications by Oostehoff et al. Adapted from Ref. [1].

**Figure 2 sensors-21-05526-f002:**
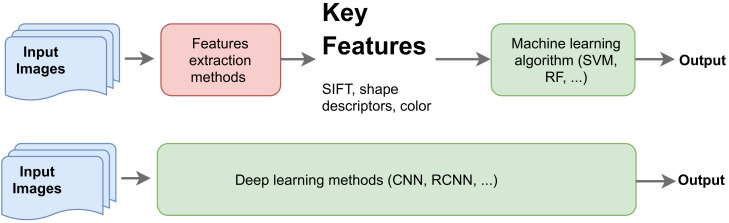
Traditional CV methods (**top**) vs. deep learning approach (**bottom**).

## Data Availability

No new data were created or analyzed in this study. Data sharing is not applicable to this article.

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
