# Peer review of "Artificial Intelligence Surgery: How Do We Get to Autonomous Actions in Surgery?"

_sensors, 2021, doi:10.3390/s21165526_

Round 1

Reviewer 1 Report

This is an interesting review paper regarding achieving autonomous robotic actions in surgery.

Authors claim that AIS involves movement. This reviewer believes that AIS involves more than movement, it involves physical human-robot interaction from the robot and the surgeon side and from the robot to the patient side.

On the one hand, stable and transparent physical human-robot interaction must be achieved from the robot-surgeon side; this might be rendered by high performance controllers and properly designed mechanical interfaces for the surgery.

On the other hand, safe physical robot-patient interaction is of paramount importance for robots to achieve autonomous actions in surgery.  How do we know  if the robot or the surgeon fails in the performance in any surgery task?. This might lead to a relevant discussion about ethics in autonomous robotic surgery.

This discussion of these topics must be reviewed by the authors in order to get more sharpen conclusions of their work.

Author Response

COMMENTS TO REVIEWER 1:

This is an interesting review paper regarding achieving autonomous robotic actions in surgery.

Authors claim that AIS involves movement. This reviewer believes that AIS involves more than movement, it involves physical human-robot interaction from the robot and the surgeon side and from the robot to the patient side.

On the one hand, stable and transparent physical human-robot interaction must be achieved from the robot-surgeon side; this might be rendered by high performance controllers and properly designed mechanical interfaces for the surgery.

On the other hand, safe physical robot-patient interaction is of paramount importance for robots to achieve autonomous actions in surgery.  How do we know  if the robot or the surgeon fails in the performance in any surgery task?. This might lead to a relevant discussion about ethics in autonomous robotic surgery.

This discussion of these topics must be reviewed by the authors in order to get more sharpen conclusions of their work.

Thank you for highlighting this important point and your extremely helpful comments. A new section on the ethics of new technological advancements in AIS was added, and we believe the paper is much improved:

“The ethics of the introduction of new surgical devices is a complex issue that has been divided into 4 broad principles: beneficence, non-maleficence, autonomy and justice[77]. In general 6 groups of people are considered when discussing this issue: the medical regulatory board, surgeons, patients, hospitals and surgical centers, industry and surgical societies[78]. The issue of ethics as it pertains to artificial intelligence surgery or more specifically autonomous actions in surgery is unclear because in general surgical robots that have already been approved for use in humans will ultimately be the vehicle used to deliver the artificially intelligent movements. Technically, the artificial intelligence will not be inside of the patient, but the robotic arms will be. Approval for technology that does not enter into the patient is generally easier to obtain then for new technology that goes inside patients. Currently examples of autonomously functioning  devices in patients include, for example, automatic staplers (iDrive, Medtronic, Dublin, Ireland). As mentioned, this device has a sensor and changes the speed of stapling and functions autonomously once activated. As autonomous actions become more complex, it begs the question of whether or not approval of more artificially intelligent devices will require a more rigorous approval process, even if the AI technical remains outside of patients. 

As opposed to the ethical issues of AI in the collection, manipulation and interpretation of medical data, AIS has the added potential danger of real-time analysis of intra-operative issues and potential for complications[79]. Alternatively, it could be argued that AI may result in fewer complications because of technology devised to minimize complications. Clearly we are many years away from being able to truly study this, nonetheless, it is clear that more surgeons need to become well-versed with issues of AI so that surgeons can truly partner with engineers and computer scientists in the development of safe autonomous actions. 

Currently, 4 classes of medical devices based on risk have been designated. Class 1 devices have minimal risk to the patient and remain in limited contact with the patient and include surgical instruments; class 2 devices include things like CT scanners, ultrasound probes and contact lenses; class 3 includes hemodialysis and implants and class 4 includes implantable defibrillators and pacemakers[80]. One wonders if a 5th class should be designated for devices that make realtime clinical decisions using AI during procedures. The consequences of not having surgeons be intimately knowledgeable and involved in the development of new technologies such as surgical robotics and AI have been divided into 5 ethical categories of experience for surgeons and additionally patient expectations and include: rescue, proximity, ordeal, aftermath and presence[81]. It is in this spirit of humility and acknowledgement of the fundamental role of morality in surgery that this article was written.”

Reviewer 2 Report

This paper is a comprehensive review paper on how automation can be achieved in the field of surgery using Artificial Intelligence (AI).

The content consists of the application of Machine Learning, Natural Language Processing (NLP), and Computer Vision (CV) with Deep Learning.

However, overall, the introduction of general machine learning and deep learning technology rather than the application of AI to the surgery field is large.

In the case of NLP, it can be used for EMR application, etc., but it has little relevance to the application of AI to achieve autonomous in the field of surgery.

The computer vision field is more related to autonomous surgery, but in 4.1 and 4.2, only general deep learning explanations and application to the overall medical field are described, and there is too little content related to autonomous surgery.

In addition, shared autonomy research based on reinforcement learning are being conducted in the surgery field, but there is no mention at all.

Therefore, as a result of comprehensively reviewing this thesis, I reject the paper approval.

Author Response

Response to REVIEWER 2:

This paper is a comprehensive review paper on how automation can be achieved in the field of surgery using Artificial Intelligence (AI).

The content consists of the application of Machine Learning, Natural Language Processing (NLP), and Computer Vision (CV) with Deep Learning.

However, overall, the introduction of general machine learning and deep learning technology rather than the application of AI to the surgery field is large.

In the case of NLP, it can be used for EMR application, etc., but it has little relevance to the application of AI to achieve autonomous in the field of surgery.

1. We agree with you, as a result, this was added to the NLP section:

“Although the technology does not currently exist for NLP to influence autonomous actions in surgery, it must be remembered that communication among team members during a surgery is fundamental to the successful performance of surgery. Additionally, devices already exist and are used today that are voice-activated (ViKY, Endocontrol, Grenoble, France)  and it is conceivable that NLP could eventually evolve to benefit the action of voice-controlled devices during a procedure[29, 30].”

The computer vision field is more related to autonomous surgery, but in 4.1 and 4.2, only general deep learning explanations and application to the overall medical field are described, and there is too little content related to autonomous surgery.

In addition, shared autonomy research based on reinforcement learning are being conducted in the surgery field, but there is no mention at all.

Therefore, as a result of comprehensively reviewing this thesis, I reject the paper approval.

2. This is an excellent point. We added a section reviewing Reinforcement learning (RL) as it pertains to autonomous actions in Artificial Intelligence Surgery. The audience that we are targeting is surgeons for this article. Our hope is that this paper can act as an introduction for surgeons to become more interested in developing autonomous action in surgery and to hopefully better embrace this evolving technology. Our plan was to write an entire paper on RL in AIS for our follow-up paper, but upon reflection, we agree that it needs to be referenced here, hence this was added to the body of the manuscript:

“4.3. Reinforcement Learning

Reinforcement learning (RL) is a branch of ML that uses Markov Decision Processes (MDP), which are based on Markov chains, named after Andrey Markov a Russian mathematician who used this framework to help predict outcomes of random events[66, 67]. In addition to AI in medicine, it has also been applied to autonomous actions in surgery[68]. As noted, enabling the computer to see and recognize things was in the past a great hindrance to advancements in AI. Ultimately an algorithm using a fusion of kinematic and video data was created based on Markov models that enabled segmentation of distinct surgical movements[69]. Combining these innovations with neural networks of DL has enabled vast improvements in the reaction time and overall speed of these complex gestures, and is known as deep reinforcement learning[69].

Currently, most research in RL and surgery involves simulators[70]. Due to the vast amount of data that must be analyzed researchers first used 2-dimensional (2D) models to devise researchable RL applications. One such model involves a 2D model for teaching gynecological surgeons how to do a hysterectomy. The researchers theorized that by defining precise goals or rewards (also known as “models”) and reproducible tasks or movements mimicking a hysterectomy that they could combine RL with surgery and develop worthwhile algorithms that can be analyzed, studied and improved[70]. 

Subsequent teams developed algorithms that were able to control robotic arms of the da Vinci system and complete tasks in 3-dimensions(3-D), specifically, identify a real 3-D cylinder with precision and accuracy and move the 3-D object[71]. Initial work was done using simulation and then transferred to real life. They also used MDPs to accomplish these tasks. Ultimately, these researchers were able to program the robot to identify  fake blood and shrapnel and to remove this from real life box trainers. These researcher ultimately made this technology open source and termed it dVRL or da Vinci Reinforcement Learning  [71].

Other groups also used RL with the da Vinci robot simulator, but referred model-free RL algorithms, or non-MDP algorithms[72]. A simple way to understand model-free RL is to think of it as learning based on “trial and error,” as opposed to other approaches that are based on “rewards.” Or, as mentioned above, “models.” A group from Johns Hopkins termed their non-model algorithm Hybrid Batch Learning (HBL) and found that they could improve the speed of their robot using this approach. They added a second replay buffer to a Q-learning process that utilized interpreted data from recorded images to stream-line decisions on movements resulting in reduced times for autonomous robotic actions. Q-learning is a form of model-free RL, and maybe the most promising form of RL for autonomous actions in surgery as it most closely resembles the human experience[72]. A Q-learning form of RL was also recently developed using the da Vinci Research Kit (Intuitive Surgical, Sunnyvale, CA, USA) to create a system to perform autonomous surgical knots using the robot[73]. However, as in current surgery what we gain in speed we often end up paying for in increased complications, which brings up the important question of ethics in AI surgical innovation.

Imitation Learning (IL) can be considered as emerging field from the domain of RL, and as a result of the recent advances in deep learning and deep convolutional neural network. The ideas is based on teaching an agent (a learning machine/ algorithm) to perform a certain task from demonstrations. The overall aim is to learn a mapping function h(x) that maps between observations and actions[74]. IL-based methods have been applied successfully to generic navigation tasks, where the agent learns from visual observations (mostly videos or images)[75]. In a recent study Kim et al. used reinforcement learning to facilitate accurate manipulation and navigation of surgical tools during eye surgery, in particular, they proposed a method to predict the relative goal position on the retinal surface from the current tool-tip position[76].”

Reviewer 3 Report

This work focuses on the discussion of the use of computational tools in the surgery field. It also brings a valuable discussion about the perception of computational assistance tools by the surgeons and medical community in general. Along with the text, the authors discuss the perception of AI by the medical community, wondering if the currently available methods, which present a high precision, can not be considered AI. This is a very pertinent and interesting discussion since the term AI has been used informally as a synonym for highly accurate algorithms. On the other hand, the term AI is also informally used as a synonym for machine learning. The boom of deep learning has demanded the implementation of machine learning solutions for a wide range of medical problems. Some of these problems were already well-investigated and present computational solutions with impressive results. As mentioned by the authors, the effort, in this case, should be for convincing surgeons to adopt the use of accurate computational tools, even if they are not based on machine learning. Moreover, the focus should be on the awareness that not only autonomous tools based on machine learning are reliable, there are plenty of non-ML-based computer vision tools that present a good accuracy and can be adopted in the surgery field, as the authors mentioned in the introduction and discussion sections.
Another important aspect discussed by the authors is the necessity of using haptics as a part of the computer vision process of autonomous robotic tools. Even considering that from a human perspective, tact is a key aspect in the tissue handling process, the way that computers and machines handle and identify different tissues may not depend on tact but on different aspects instead. More investigations in this field may reveal different data that can be used for "machine tissue vision" that may be more precise than the current human-inspired tact. This discussion was brought in a very appropriate way by the authors.
The authors also exposed a major challenge in the adoption of computational methods, especially the ones based on supervised machine learning, which is the high variability of cases and the lack of annotations. For the specific field of surgery, this can be an even more serious concern since data is even rarer when compared with other medical fields. 

Some suggestions to improve the manuscript quality are listed below:

Although the abstract suggests that this a review article, this work is more of a comparative discussion about the advancements of computational methods and tools for two different medical fields: surgery and diagnosis. To present this work as a review may create an expectation for the readers that the work is a formal literature review, which is not the case. Please, consider altering the abstract to better express the main objective of the work.

The medical community's resistance to adopt computational methods to automate tasks and processes was observed before the recent advances of machine learning, as discussed by Randolph et al. ("How to Use an Article Evaluating the Clinical Impact of a Computer-Based Clinical Decision Support System", doi:10.1001/jama.282.1.67 ). Please consider including some statements about this in the introduction section.

It is essential to formally include some of the mentioned concepts to avoid any misunderstanding, especially in the introduction section. Some of the mentioned tools are considered as "Weak AI". Nevertheless, this term is not in the common domain. Therefore, it should be better explained by the authors. 

In the discussion section, the authors discuss an ethical concern about using AI in surgery: "Although an automated surgical robot should ultimately have the ability to act with intention, it is hard to see the need for a robot having beliefs or desires to function effectively during an interventional procedure". It is important to highlight that the huge majority of AI systems used nowadays are focused on specific tasks and specific scenarios. In that way, this concern is much more theoric than practical, i.e., it considers advances in AI that may be achieved in the long term future, as recently comment by the Turing awardees Yoshua Bengio, Yann Lecun, Geoffrey Hinton in their recent lecture (Deep Learning for AI, doi: 10.1145/3448250, https://cacm.acm.org/magazines/2021/7/253464-deep-learning-for-ai/fulltext).
In other words, even considering that this ethical concern is valid, it is important to mention that it is focused on possible future advances and not on the majority of current AI medical technologies. Nevertheless, as the authors mentioned, there is a need for ethical standards for the use of autonomous tools, especially for the cases in which failures caused by the autonomous tools inflict patients. Should the surgeons legally respond for this?

In the abstract and introduction section, it is implicit that this manuscript would focus on the use of AI for autonomous robotics in surgery as a tool to be used during the surgical intervention itself. Nevertheless, as is explained by the authors in the following sections, AI can be employed in other aspects of surgery, as pre-operative and post-operative.
In that way, the authors should evaluate and comment if the term AIS covers only autonomous surgery supported by AI or if it also includes the use of AI in other surgery-related processes, as pre and post-operative assessments. Please, add some statements in the abstract and the introduction mentioning pre and post-operative as other aspects that can include AI tools.

Author Response

RESPONSE TO REVIEWER 3:

This work focuses on the discussion of the use of computational tools in the surgery field. It also brings a valuable discussion about the perception of computational assistance tools by the surgeons and medical community in general. Along with the text, the authors discuss the perception of AI by the medical community, wondering if the currently available methods, which present a high precision, can not be considered AI. This is a very pertinent and interesting discussion since the term AI has been used informally as a synonym for highly accurate algorithms. On the other hand, the term AI is also informally used as a synonym for machine learning. The boom of deep learning has demanded the implementation of machine learning solutions for a wide range of medical problems. Some of these problems were already well-investigated and present computational solutions with impressive results. As mentioned by the authors, the effort, in this case, should be for convincing surgeons to adopt the use of accurate computational tools, even if they are not based on machine learning. Moreover, the focus should be on the awareness that not only autonomous tools based on machine learning are reliable, there are plenty of non-ML-based computer vision tools that present a good accuracy and can be adopted in the surgery field, as the authors mentioned in the introduction and discussion sections.

Another important aspect discussed by the authors is the necessity of using haptics as a part of the computer vision process of autonomous robotic tools. Even considering that from a human perspective, tact is a key aspect in the tissue handling process, the way that computers and machines handle and identify different tissues may not depend on tact but on different aspects instead. More investigations in this field may reveal different data that can be used for "machine tissue vision" that may be more precise than the current human-inspired tact. This discussion was brought in a very appropriate way by the authors.

The authors also exposed a major challenge in the adoption of computational methods, especially the ones based on supervised machine learning, which is the high variability of cases and the lack of annotations. For the specific field of surgery, this can be an even more serious concern since data is even rarer when compared with other medical fields. 

THANK YOU for your wonderful review and suggestions! We really can’t thank you enough for the time and care that you took in reviewing our article. We did our best address all of your suggestions, if we missed anything please let us know.

Some suggestions to improve the manuscript quality are listed below:

Although the abstract suggests that this a review article, this work is more of a comparative discussion about the advancements of computational methods and tools for two different medical fields: surgery and diagnosis. To present this work as a review may create an expectation for the readers that the work is a formal literature review, which is not the case. Please, consider altering the abstract to better express the main objective of the work.

1. We agree that this has rendered the manuscript less clear. As a result we better clarified the aims of this paper and removed any reference to a literature review.

The last lines to the abstract were changed to:

This article will discuss aspects of ML, DL, CV and NLP as they pertain to the modern practice of surgery with a focus on current AI issues and advances that will enable us to get to more autonomous actions in surgery.  Ultimately, there may be a paradigm shift that needs to occur in the surgical community as more surgeons with expertise in AI may be needed to fully unlock the potential of AIS in a safe, efficacious and timely manner.

2. In addition the last line of the introduction was changed to:

This manuscript will discuss the literature on ML, DL, NLP and CV as it pertains to AIM for the pre-operative diagnosis and post-operative management of surgical patients and autonomous robotics in surgery, to attempt to ascertain the current obstacles and next steps necessary in the evolution towards AI/autonomous surgery.

The medical community's resistance to adopt computational methods to automate tasks and processes was observed before the recent advances of machine learning, as discussed by Randolph et al. ("How to Use an Article Evaluating the Clinical Impact of a Computer-Based Clinical Decision Support System", doi:10.1001/jama.282.1.67 ). Please consider including some statements about this in the introduction section.

3. This is a great reference, thank you. This section now reads:

Unfortunately, the reluctance of many laparoscopic surgeons to give up on haptics, or the sense of touch, is actually hindering progress in AIS because of the refusal to embrace robotic telemanipulation technology, in effect, they are refusing to let go, something that will be needed if the dream of AIS is ever to come to pass(Gumbs, De Simone et al. 2020). Unfortunately, the medical community has already been shown to be resistant to any automation of medical tasks even simple computations, unfortunately it is safe to say that automation of surgical tasks will have an even more profound degree of resistance (Randolph, Haynes et al. 1999).

It is essential to formally include some of the mentioned concepts to avoid any misunderstanding, especially in the introduction section. Some of the mentioned tools are considered as "Weak AI". Nevertheless, this term is not in the common domain. Therefore, it should be better explained by the authors. 

4. This was added to the Discussion to improve the explanation:

Essentially, “Weak” AI is an automatic action, but without the idealized concept of intelligence that one would expect in AI, specifically, a complex interpretation of a problem with an advanced solution equal or superior to a humans.

In the discussion section, the authors discuss an ethical concern about using AI in surgery: "Although an automated surgical robot should ultimately have the ability to act with intention, it is hard to see the need for a robot having beliefs or desires to function effectively during an interventional procedure". It is important to highlight that the huge majority of AI systems used nowadays are focused on specific tasks and specific scenarios. In that way, this concern is much more theoric than practical, i.e., it considers advances in AI that may be achieved in the long term future, as recently comment by the Turing awardees Yoshua Bengio, Yann Lecun, Geoffrey Hinton in their recent lecture (Deep Learning for AI, doi: 10.1145/3448250, https://cacm.acm.org/magazines/2021/7/253464-deep-learning-for-ai/fulltext).

5. We understand and agree with your point, as a result, this statement was modified to:

Although an automated surgical robot should ultimately have the ability to act with intention, it is hard to see the need for a robot having beliefs or desires to function effectively during an interventional procedure, however, this may not be the case in the future once AI is much more advanced(Bengio, Lecun et al. 2021).

In other words, even considering that this ethical concern is valid, it is important to mention that it is focused on possible future advances and not on the majority of current AI medical technologies. Nevertheless, as the authors mentioned, there is a need for ethical standards for the use of autonomous tools, especially for the cases in which failures caused by the autonomous tools inflict patients. Should the surgeons legally respond for this?

6. The ethical considerations for the adoption of autonomous actions in surgery are complex. Honestly, we don’t really know right now if surgeons should be responsible for complications from AIS. Regardless, in an attempt to highlight this issue we added a comprehensive section discussing the ethics of technological advances in AIS:

The ethics of the introduction of new surgical devices is a complex issue that has been divided into 4 broad principles: beneficence, non-maleficence, autonomy and justice[77]. In general 6 groups of people are considered when discussing this issue: the medical regulatory board, surgeons, patients, hospitals and surgical centers, industry and surgical societies[78]. The issue of ethics as it pertains to artificial intelligence surgery or more specifically autonomous actions in surgery is unclear because in general surgical robots that have already been approved for use in humans will ultimately be the vehicle used to deliver the artificially intelligent movements. Technically, the artificial intelligence will not be inside of the patient, but the robotic arms will be. Approval for technology that does not enter into the patient is generally easier to obtain then for new technology that goes inside patients. Currently examples of autonomously functioning  devices in patients include, for example, automatic staplers (iDrive, Medtronic, Dublin, Ireland). As mentioned, this device has a sensor and changes the speed of stapling and functions autonomously once activated. As autonomous actions become more complex, it begs the question of whether or not approval of more artificially intelligent devices will require a more rigorous approval process, even if the AI technical remains outside of patients. 

As opposed to the ethical issues of AI in the collection, manipulation and interpretation of medical data, AIS has the added potential danger of real-time analysis of intra-operative issues and potential for complications[79]. Alternatively, it could be argued that AI may result in fewer complications because of technology devised to minimize complications. Clearly we are many years away from being able to truly study this, nonetheless, it is clear that more surgeons need to become well-versed with issues of AI so that surgeons can truly partner with engineers and computer scientists in the development of safe autonomous actions. 

Currently, 4 classes of medical devices based on risk have been designated. Class 1 devices have minimal risk to the patient and remain in limited contact with the patient and include surgical instruments; class 2 devices include things like CT scanners, ultrasound probes and contact lenses; class 3 includes hemodialysis and implants and class 4 includes implantable defibrillators and pacemakers[80]. One wonders if a 5th class should be designated for devices that make realtime clinical decisions using AI during procedures. The consequences of not having surgeons be intimately knowledgeable and involved in the development of new technologies such as surgical robotics and AI have been divided into 5 ethical categories of experience for surgeons and additionally patient expectations and include: rescue, proximity, ordeal, aftermath and presence[81]. It is in this spirit of humility and acknowledgement of the fundamental role of morality in surgery that this article was written.

In the abstract and introduction section, it is implicit that this manuscript would focus on the use of AI for autonomous robotics in surgery as a tool to be used during the surgical intervention itself. Nevertheless, as is explained by the authors in the following sections, AI can be employed in other aspects of surgery, as pre-operative and post-operative.
In that way, the authors should evaluate and comment if the term AIS covers only autonomous surgery supported by AI or if it also includes the use of AI in other surgery-related processes, as pre and post-operative assessments. Please, add some statements in the abstract and the introduction mentioning pre and post-operative as other aspects that can include AI tools.

7. As mentioned, this was added to the ABSTRACT to clarify this:

This article will discuss aspects of ML, DL, CV and NLP as they pertain to the modern practice of surgery with a focus on current AI issues and advances that will enable us to get to more autonomous actions in surgery.

And this was adde dto the end of the introduction:

This manuscript will discuss the literature on ML, DL, NLP and CV as it pertains to AIM for the pre-operative diagnosis and post-operative management of surgical patients and autonomous robotics in surgery, to attempt to ascertain the current obstacles and next steps necessary in the evolution towards AI/autonomous surgery.

Round 2

Reviewer 2 Report

Overall, the cases and contents of applying AI to the surgery field have been supplemented, but the contents in the computer vision field are still poor. This paper introduces basic methods of CNN-based object localization and detection and cases applied to various medical image modalities. However, since there is no information introduced in the surgery field, it is recommended to add cases where methods such as object localization and detection are applied.

For example, studies that applied classification, detection, tracking, segmentation, and phase recognition to laparoscopic and robotic surgery images are as follows.

I recommend you to cite these papers to enhance the content and improve the completeness.

tool classification

Twinanda, Andru P., et al. "Endonet: a deep architecture for recognition tasks on laparoscopic videos." IEEE transactions on medical imaging 36.1 (2016): 86-97.

Sahu, Manish, et al. "Addressing multi-label imbalance problem of surgical tool detection using CNN." International journal of computer assisted radiology and surgery 12.6 (2017): 1013-1020.

Mishra, Kaustuv, Rachana Sathish, and Debdoot Sheet. "Learning latent temporal connectionism of deep residual visual abstractions for identifying surgical tools in laparoscopy procedures." Proceedings of the IEEE Conference on Computer Vision and Pattern Recognition Workshops. 2017.

tool detection and tracking

Sarikaya, Duygu, Jason J. Corso, and Khurshid A. Guru. "Detection and localization of robotic tools in robot-assisted surgery videos using deep neural networks for region proposal and detection." IEEE transactions on medical imaging 36.7 (2017): 1542-1549.

Sarikaya, Duygu, Jason J. Corso, and Khurshid A. Guru. "Detection and localization of robotic tools in robot-assisted surgery videos using deep neural networks for region proposal and detection." IEEE transactions on medical imaging 36.7 (2017): 1542-1549.

Lee, Dongheon, et al. "Evaluation of Surgical Skills during Robotic Surgery by Deep Learning-Based Multiple Surgical Instrument Tracking in Training and Actual Operations." Journal of clinical medicine 9.6 (2020): 1964.

tool segmentation

Shvets, Alexey A., et al. "Automatic instrument segmentation in robot-assisted surgery using deep learning." 2018 17th IEEE International Conference on Machine Learning and Applications (ICMLA). IEEE, 2018.

Marban, Arturo, et al. "Estimating position & velocity in 3d space from monocular video sequences using a deep neural network." Proceedings of the IEEE International Conference on Computer Vision Workshops. 2017.

García-Peraza-Herrera, Luis C., et al. "Real-time segmentation of non-rigid surgical tools based on deep learning and tracking." International Workshop on Computer-Assisted and Robotic Endoscopy. Springer, Cham, 2016.

phase recognition

Yu, Felix, et al. "Assessment of automated identification of phases in videos of cataract surgery using machine learning and deep learning techniques." JAMA network open 2.4 (2019): e191860-e191860.

Khalid, Shuja, et al. "Evaluation of deep learning models for identifying surgical actions and measuring performance." JAMA network open 3.3 (2020): e201664-e201664.

Author Response

Overall, the cases and contents of applying AI to the surgery field have been supplemented, but the contents in the computer vision field are still poor. This paper introduces basic methods of CNN-based object localization and detection and cases applied to various medical image modalities. However, since there is no information introduced in the surgery field, it is recommended to add cases where methods such as object localization and detection are applied.

For example, studies that applied classification, detection, tracking, segmentation, and phase recognition to laparoscopic and robotic surgery images are as follows.

I recommend you to cite these papers to enhance the content and improve the completeness.

tool classification

Twinanda, Andru P., et al. "Endonet: a deep architecture for recognition tasks on laparoscopic videos." IEEE transactions on medical imaging 36.1 (2016): 86-97.

Sahu, Manish, et al. "Addressing multi-label imbalance problem of surgical tool detection using CNN." International journal of computer assisted radiology and surgery 12.6 (2017): 1013-1020.

Mishra, Kaustuv, Rachana Sathish, and Debdoot Sheet. "Learning latent temporal connectionism of deep residual visual abstractions for identifying surgical tools in laparoscopy procedures." Proceedings of the IEEE Conference on Computer Vision and Pattern Recognition Workshops. 2017.

tool detection and tracking

Sarikaya, Duygu, Jason J. Corso, and Khurshid A. Guru. "Detection and localization of robotic tools in robot-assisted surgery videos using deep neural networks for region proposal and detection." IEEE transactions on medical imaging 36.7 (2017): 1542-1549.

Sarikaya, Duygu, Jason J. Corso, and Khurshid A. Guru. "Detection and localization of robotic tools in robot-assisted surgery videos using deep neural networks for region proposal and detection." IEEE transactions on medical imaging 36.7 (2017): 1542-1549.

Lee, Dongheon, et al. "Evaluation of Surgical Skills during Robotic Surgery by Deep Learning-Based Multiple Surgical Instrument Tracking in Training and Actual Operations." Journal of clinical medicine 9.6 (2020): 1964.

tool segmentation

Shvets, Alexey A., et al. "Automatic instrument segmentation in robot-assisted surgery using deep learning." 2018 17th IEEE International Conference on Machine Learning and Applications (ICMLA). IEEE, 2018.

Marban, Arturo, et al. "Estimating position & velocity in 3d space from monocular video sequences using a deep neural network." Proceedings of the IEEE International Conference on Computer Vision Workshops. 2017.

García-Peraza-Herrera, Luis C., et al. "Real-time segmentation of non-rigid surgical tools based on deep learning and tracking." International Workshop on Computer-Assisted and Robotic Endoscopy. Springer, Cham, 2016.

phase recognition

Yu, Felix, et al. "Assessment of automated identification of phases in videos of cataract surgery using machine learning and deep learning techniques." JAMA network open 2.4 (2019): e191860-e191860.

Khalid, Shuja, et al. "Evaluation of deep learning models for identifying surgical actions and measuring performance." JAMA network open 3.3 (2020): e201664-e201664.

Reply:

Thank you for your detailed comments. The entire article was again re-read and edited by the first author who despite having moved to France in 2018 is a native english-speaker with a British Passport born and raised in the United States of America. The english hopefully flows better now.

As for the comments on the Computer Vision section, an entire section was added before the Reinforcement Learning section and is attached below (4.3.CV in AIS). All requested articles were cited.

4.3. CV in AIS

The significant progress that took place in objects recognition and localization in 2D and 3D images have been reflected in autonomous surgery across different types of applications such as phase recognition[1-3], detection and tracking of objects of interest [4-6] and segmentation[7, 8]. Phase recognition is an important aspect for training and educating of doctors by using videos of various types of surgery. However, despite the availability of these videos their use in training is still limited, because these videos require some sort of pre-processing, and also segmentation into different phases for subsequent automated skill assessment and feedback[3].

To address this issue, Twinanda et al.[1] build a CNN-based method to perform phase recognition in laparoscopic surgery directly from raw pixels (image and videos). The authors used a dataset of 80 videos of cholecystectomies performed by 13 surgeons to train the CNN, and promising results were reported in terms of the model's ability to handle complex scenarios and outperform other traditional tools. Similarly, Yu et al.[2] used five different algorithms including CNN for handling the of videos of cataract surgery and Recurrent Neural Networks (RNN) for handling time series data with labels. Results clearly showed that deep learning techniques (CNN and RNN) provide better options for learning from time series data and video images, and can provide accurate and automated detection of phases in cataract surgery.  Khalid et all.[3] utilized deep convolutional neural networks with embedding representation for phase recognition. The authors used 103 video clips of table-top surgical procedures, performed by 8 surgeons, including 4 to 5 trials of 3 surgical actions. Promising results using precision, recall and accuracy were reported in terms of the model’s ability to classify performance level and surgical actions.

Object detection and tracking is another important aspect of AI in surgery that has progressed due to the latest developments in deep learning and deep convolutional neural networks. Sarikaya et al.[5] used a dataset of videos from 10 surgeons and applied a deep convolutional neural network to speed up detection and localization of instruments in robot-assisted surgery (RAS). The authors used multimodal CNNs to capture objects of interest and the temporal aspect of the data (motion cues). Results with 91% precision were reported and with relatively good performance in terms of computational time (0.1 seconds per frame).

Tracking of objects of interest across different frames is another key aspect of AIS that has also advanced due to the latest developments in computer vision. Lee et al.[6] proposed a deep learning-based method for tracking surgical instruments to evaluate surgeon’s skills in performing procedures by robotic surgery. The authors used 54 videos for training their models and used mean square root error and the area under the curve for evaluation purposes. Results showed that the proposed method was able to accurately track instruments during robotic surgery. The authors concluded that the results suggest that the current method of surgical skill assessment by surgeons can be replaced by this proposed method.

One particular application of CV that has seen significant progress in recent years due to developments of deep learning-based methods, is image and video segmentation. Accurate segmentation of images and videos is crucial for AIS and robot-assisted surgery. A notable example from the Massachusetts Institute of Technology proposed a deep learning-based approach for robotic instrument segmentation[8]. The authors proposed an architecture based on deep residual models (U-Net)[9]. The method presented is providing pixel-level segmentation, where each pixel in the image/video is labelled as an instrument or background, and the authors used 8 videos (each one is 255 frames) for training their models, and reported comparable results with the state-of-the-art. Similarly, in [7] the authors, used fully convolutional network and optical tracking for segmentation of computer-assisted surgery videos. Overall results of 80.6% of balanced accuracy was reported in a non-real time version of the method, and dropped to 78.2% balanced accuracy for the real-time version.

Various other applications of CV methods and AI can be seen in AIS, including application of CV and AI for education in surgery [10], improves efficiency in the operating room [11], during neurosurgery [12], and other surgical disciplines. For a comprehensive and recent review of the use of computer vision-based methods in AIS and assisted surgery, the reader is referred here [13].

Bibliography

1. Twinanda, A., et al. En- donet: A deep architecture for recognition tasks on laparoscopic videos. IEEE Transactions on Medical Imaging, 2017. 36, 86-97.

2. Yu, F., et al., Assessment of automated identification of phases in videos of cataract surgery using machine learning and deep learning techniques. JAMA Network Open, 2019. 2(4): p. e191860.

3. Khalid, S., et al., Evaluation of Deep Learning Models for Identifying Surgical Actions and Measuring Performance. JAMA Netw Open, 2020. 3(3): p. e201664.

4. Marban, A., et al. Estimating position amp; velocity in 3d space from monocular video sequences using a deep neural network. in IEEE International Conference on Computer Vision Workshops (ICCVW). 2017.

5. Sarikaya, D., J. Corso, and K. Guru, Detection and localization of robotic tools in robot- assisted surgery videos using deep neural networks for region proposal and detection. IEEE Transactions on Medical Imaging, 2017. 36(7 1): p. 1542–1549.

6. Lee, D., et al., Evaluation of surgical skills during robotic surgery by deep learning-based multiple surgical instrument tracking in training and actual operations. Journal of Clinical Medicine, 2020. 9(6): p. 1964.

7. Garcıa-Peraza-Herrera, L., et al., Real-time segmentation of non-rigid surgical tools based on deep learning and tracking, in Computer-Assisted and Robotic Endoscopy, T. Peters, et al., Editors. 2017, Springer International Publishing.

8. Shvets, A., et al. Automatic instrument seg- mentation in robot-assisted surgery using deep learning. in 17th IEEE International Conference on Machine Learning and Applications (ICMLA). 2018.

9. He, K., et al. Deep residual learning for image recognition. in Proceedings of the IEEE Conference on Computer Vision and Pattern Recognition (CVPR). 2016.

10. Ward, T., et al., Surgical data science and artificial intelligence for surgical education. Journal of Surgical Oncology, 2021. 124(2): p. 221–230.

11. Birkhoff, D., A. van Dalen, and M. Schijven, A review on the current applications of artificial intelligence in the operating room. Surgical Innovation, 2021.

12. Pangal, D., et al., A Guide to Annotation of Neurosurgical Intraoperative Video for Machine Learning Analysis and Computer Vision,. World Neurosurgery, 2021. 150: p. 26-30.

13. Kennedy-Metz, L., et al. Computer vision in the operating room: Opportunities and caveats. in IEEE Transactions on Medical Robotics and Bionics. 2021.